# LRRK2 Targeting Strategies as Potential Treatment of Parkinson’s Disease

**DOI:** 10.3390/biom11081101

**Published:** 2021-07-26

**Authors:** Dominika Natalia Wojewska, Arjan Kortholt

**Affiliations:** 1Faculty of Science and Engineering, University of Groningen, Nijenborg 7, 9747AG Groningen, The Netherlands; d.n.wojewska@gmail.com; 2YETEM-Innovative Technologies Application and Research Center, Suleyman Demirel University, 32260 Isparta, Turkey

**Keywords:** kinase inhibitors, neurodegenerative diseases, Parkinson’s disease, protein–protein interactions, small GTPases, LRRK2

## Abstract

Parkinson’s Disease (PD) affects millions of people worldwide with no cure to halt the progress of the disease. Leucine-rich repeat kinase 2 (LRRK2) is the most common genetic cause of PD and, as such, LRRK2 inhibitors are promising therapeutic agents. In the last decade, great progress in the LRRK2 field has been made. This review provides a comprehensive overview of the current state of the art, presenting recent developments and challenges in developing LRRK2 inhibitors, and discussing extensively the potential targeting strategies from the protein perspective. As currently there are three LRRK2-targeting agents in clinical trials, more developments are predicted in the upcoming years.

## 1. Introduction

Neurodegeneration is defined as age-related, progressive loss of structure and function of neurons, ultimately leading to their death. This in turn leads to severe impairment of many crucial brain functions, such as memory loss, personality changes, and impaired mobility. Because of the ever increasing life span, neurodegenerative diseases pose a major medical challenge worldwide [1].

Parkinson’s disease (PD) is the second most common neurodegenerative disorder, after Alzheimer’s disease. Currently available PD treatments are only symptomatic, with no cure that would stop the progress of the disease or reverse it [2,3]. Early onset diagnosis would allow for better control of the symptoms; however, the current diagnostic toolset relies on presenting them, which is already a sign of advanced neurodegeneration [2,4]. At the moment, there are no specific tests or biomarkers in clinical use that would allow for detecting the disease at an early stage, before presenting the symptoms, although there are some markers that could give an indication for PD [2,5]. PD diagnosis and treatment are further complicated by the various forms of disease due to different underlying causes, which in turn render some of the available treatments ineffective [3,6]. Only about 5–15% of PD cases have a familial history of the disease [4], with most of the cases being idiopathic.

Epidemiological studies have shown that there is strong genetic correlation, with mutations in six genes being the major cause: *snca* (α-synuclein), *lrrk2* (Leucine-rich repeat kinase 2), *vps35* (Vacuolar protein sorting ortholog 35), *prkn* (Parkin), *dj1* (DJ-1), and *pink1* (PTEN-induced kinase 1) [7,8].

The age of PD onset varies significantly for those genes, and so does the type of inheritance, which can be either dominant (*snca*, *lrrk2*, *vps35*) or recessive (*prkn*, *dj-1* and *pink1*). Furthermore, recent genome-wide association studies (GWAS) have shown that there is high genetic heterogeneity at the basis of PD, with great racial difference of the genetic causes in different populations and many rare mutations are usually detected only in a single family or in small populations. Among all identified risk-genes, mutations in *lrrk2* stand out as the most common, single genetic cause of PD. The most common *lrrk2* G2019S mutation may explain up to 5% of all PD cases and is especially high among Ashkenazi Jews and North African Arabs [9]. Importantly, recent studies have also detected an increased LRRK2 activity in post-mortem brain tissue from patients with idiopathic disease [10]. Therefore, LRRK2 has become an essential player in PD pathogenesis, both sporadic and familiar, and insight into LRRK2 function may thus help to understand the progression of PD in general. Hence, targeting LRRK2 might not only be important for LRRK2-mediated PD, but also has the potential to address PD caused by other gene mutations or agents. In this review, we will give an overview of all the various LRRK2 targeting strategies, discuss their current use, and give some future perspective on targeting PD in the clinic.

## 2. LRRK2: Structure, Function, and Role in PD

LRRK2 is a large (2527 amino acids, 286 kDa), multidomain protein that bears two enzymatic functions: kinase and GTPase, and several protein–protein interaction (PPI) domains. There are many genomic PD variants of LRRK2, some of them have been repeatedly confirmed as pathogenic, others are considered risk factors, and some are benign [7]. Figure 1a depicts the domain structure of LRRK2 and the location of the seven PD-pathological mutations. Several other mutations in LRRK2 have been linked to other diseases, such as Crohn’s or Hansen’s disease [11,12,13].

LRRK2 is expressed in multiple tissues, with the highest expression levels in leukocytes, including monocytes, B lymphocytes, dendritic cells, and microglia [14,15]. Moreover, it is expressed in the lung and kidney, although at lower levels [16], with surprisingly low expression in the brain [17]. Despite numerous efforts, the exact physiological function of LRRK2 remains elusive, although there is a good body of evidence showing its involvement in multiple cellular processes, such as: neuronal plasticity, vesicle trafficking, mitochondrial function, autophagy, apoptosis, and many others [18]. Physiological substrates of LRRK2 kinase belong to the Rab GTPase family [19,20,21]. It has been proposed that LRRK2 might serve different functions in different tissues, depending on the subset of downstream Rab substrates being expressed.

Accumulating evidence suggests that this LRRK2/Rab pathway functions at the interface of vesicular trafficking, lysosomal functioning, and autophagy [22,23,24]. These processes play a crucial role during immune response and several studies have linked LRRK2 signaling to inflammation in general and neuroinflammation specifically (reviewed in [25]. Interestingly, recently, it was shown that LRRK1, the human homolog of LRRK2, phosphorylates a different subset of Rab proteins than LRRK2, suggesting that LRRK enzymes are Rab specific kinases [26].

### 2.1. LRRK2 Structure

Even though elucidating the structure of LRRK2 has been the goal of many experimental efforts for the last decade, the results obtained so far are still far from ideal. The main roadblock here is the stability and solubility of the purified protein. In a recent study, Deniston et al. [27] has systematically scanned for the most optimal LRRK2 construct(s) expressed in the insect cells, which led to obtaining a soluble catalytic half of LRRK2 (RocCOR-kinase-WD40) and yielded a structure with 3.5 Å resolution. This is a significant progress compared with the previous structures of full-length LRRK2 (22 Å [28], 16 Å [29], and 14 Å [30]. During the revision of this manuscript, the first high-resolution Cryo-EM structure of full-length LRRK2 was published [31]. From these studies, it is clear that LRRK2 forms tightly packed homodimers in head-to tail fashion, with a multitude of tight interactions between spatially distant domains [27,28], Figure 1b,c shows the molecular model obtained from the electron density map; domains were color-coded for clarity (PDB ID 6VNO). The RocCOR module is located at the core of the complex, which is the main dimerization interface. The N-terminus of the protein is folding back on the kinase domain, which is exposed on the opposite sides of the complex (Figure 1b). Intriguingly, the C-terminal helix that follows the WD40 domain docks onto the kinase domain at a putative allosteric site. This interaction may explain the observed stimulating effect of the WD40 domain on LRRK2 neurotoxicity [32,33]. The close proximity of the kinase domain and RocCOR supradomain from the other protomer creates room for speculation on the molecular mechanism of how these two domains regulate each other [34,35].

Although the recent EM structures are a major breakthrough in understanding the structure and activation mechanism of LRRK2, structure-based drug design efforts require high-resolution structures (namely, below 2 Å). Crystallizing the full-length or truncated LRRK2 failed to yield a structure so far, even under microgravity conditions [36]. Therefore, the structural efforts have turned into individual domains. Up to date, the amount of structural information on LRRK2 domains is limited (WD40 solved at 2.6 Å [37] and the Roc domain as swapped dimer, solved at 2.0 Å [38], later revised to 1.6 Å [39], which is believed to not be biologically relevant). The major challenge in working with other LRRK2 domains is their low solubility and issues with purification [40,41,42].

As the experimental work to obtain a high-resolution structure of individual LRRK2 domains is ongoing [38,39,43], several attempts to generate in silico models have been made. In particular, homology models of each LRRK2 domain have been published [44,45].

### 2.2. LRRK2 Activation and Cycle

Several structures of LRRK2 domains from orthologous proteins have been published, namely: RocCOR and LRR-RocCOR domains from *C. tepidum* [46] and [47], respectively; RocCOR from *M. barkeri* [48], and the kinase domain from *D. discodeum* [49]. These studies have shed some light on the probable mechanism of LRRK2 activation cycle [50,51,52] and were useful in characterizing LRRK2 inhibitors [53]. This section describes the most pliable mechanism of LRRK2 activation and cycle, which is schematically shown in Figure 2.

It has been shown that the monomeric LRRK2 is a predominantly cytosolic protein, while its homodimeric form associates onto biological membranes [54]. The GTP-bound monomeric LRRK2 state is stabilized by 14-3-3 proteins, which bind to LRRK2 upon phosphorylation of serine 910 and serine 935, thereby preventing LRRK2 aggregation in cytosolic inclusion pools [55,56]. In this stabilized monomeric state, both the GTPase and the kinase presumably only have low basal activity [54].

Accumulating evidence points to GTP-bound Rab family of GTPases as the main membrane recruiting factors [23]. A subset of Rab proteins can bind to the N-terminus of LRRK2 and thereby induce localization of LRRK2 to various membranous organelles, depending on the Rab isoform. Rab29 recruits LRRK2 to the lysosomes and Golgi, while Rab32 recruits LRRK2 to lysosomes [35]. However, a recent study has shown that a Rab29 knockout cell line still has normal LRRK2 activation [57]; therefore, further research is necessary to understand how and which Rab GTPases are functioning upstream from LRRK2. The membrane-associated LRRK2 dimers have higher kinase activity and can, subsequently, phosphorylate other Rab proteins [54,58].

Membrane association, but not the identity of the membrane, is important for LRRK2 activation, suggesting that membrane binding induces conformational changes in LRRK2 that allow for activating its kinase domain [58]. At the membrane, LRRK2 goes through a multi-step hydrolysis cycle, resulting in a dimeric, GDP-bound conformation of LRRK2 at the membrane (discussed in detail in [52]).

Initially, it was postulated that LRRK2 acts via a similar mechanism to G-proteins activated by dimerization [46,59,60], but, in the light of newer evidence, this view was abandoned [50,52], and, currently, it is believed that the activation mechanism of LRRK2 is unique. It is unclear during which step dimerization occurs; however, it is mediated via the COR domain and is crucial for hydrolysis, maximum kinase activity, and LRRK2 functioning (see Figure 2).

Studies in vitro on LRRK2 orthologs have shown that the RocCOR module stays dimeric when it is GDP-bound and nucleotide free, but it quickly monomerizes when exposed to GTP [50]. Since the GDP affinity towards Roc is low (in the micromolar range), the GTP off rate is fast, and the cellular abundance of GTP is higher than GDP, most likely, the GDP/GTP nucleotide exchange is rapid and concerted with monomerization. Interestingly, Wauters et al. showed that the Michaelis constant (*K*_M_) of the GTPase reaction is within the range of the cellular GTP concentration [52], which suggests that cellular fluxes of GTP could control the LRRK2 GTP/GDP cycle. 

So far, it remains to be determined whether the LRRK2 monomerization is sufficient to induce dissociation from the membrane, or if additional factors are needed to regulate this process [52]. After the protein monomerizes, and dissociates from the membrane, it is ready to start another cycle.

### 2.3. LRRK2 Activation in Parkinson’s Disease

The exact role LRRK2 plays in PD has not been clearly established yet. The influence of the PD-causing mutations on the LRRK2 activation mechanism and its interactors is also not clearly understood [61]. There are numerous single nucleotide substitutions in the LRRK2 gene (an extensive list can be found in the UniProt database (ID: Q5S007) or in [7]), although, for most of them, there is not enough evidence to be associated with PD or other diseases. Table 1 summarizes the clearly PD-pathogenic LRRK2 variants and their influence on LRRK2 function. Some of the mutations have been reported to have a contradictory effect on the LRRK2 kinase activity (e.g., I2020T) [62,63]. It is worth noting that each PD mutant in LRRK2 results in unique neuropathology [64], and their prevalence is specific to certain populations or even families [7]. 

Because the LRRK2 activation mechanism is complex and is regulated at several levels (see above), one can easily imagine that any amino acid substitution affecting either the enzymatic core or any of the PPI interfaces may perturb it. Based on experimental evidence, it appears that indeed this is the case, with various mutations resulting in different defects in the activation mechanism. The common mechanistic output though is increased kinase activity and lowered GTPase activity, which altogether results in prolonged kinase-active, GTP-bound (transition) state, leading to increased LRRK2 signaling, promoting neuronal cell death and ultimately causing PD [35]. Interestingly, the symptoms of LRRK2-related PD and sporadic PD are very similar and, recently, it has been found that LRRK2 activity was enhanced in postmortem brain tissue from patients with idiopathic PD [10,75]. This suggests that LRRK2 activity plays a role in PD independence of mutations and that targeting of LRRK2 thus might be beneficial for both LRRK2 carriers and for the treatment of sporadic PD.

## 3. Modes of LRRK2 Inhibition

### 3.1. Kinase Inhibitors

Kinase inhibition is a very common therapeutic strategy [76,77], and many tools for studying and fine-tuning kinase inhibitors have already been developed. Moreover, the kinase function appears to be the final outcome of LRRK2 signaling and is upregulated in all PD-causing mutants; therefore, from a pharmacological standpoint, this should be the most straightforward option to clinically target LRRK2. Numerous LRRK2 kinase inhibitors have been developed, all ATP-competitive (reviewed in [78]). It has been shown that inhibiting the kinase domain of LRRK2 has neuroprotective effects (reviewed in [17]) and prevents endolysosomal deficits [79], making it a very attractive treatment strategy. Table 2 presents some of the commercially available LRRK2 kinase inhibitors.

MLi-2, developed by MERCK in 2015, is commonly used in academics [80,81]. MLi-2 is a highly potent, very selective, ATP-competitive LRRK2 inhibitor. It was discovered in a high-throughput screening effort, and subsequently optimized to result in IC_50_ of 0.76 nM (in vitro) and 1.4 nM (in vivo), with similar values for the G2019S mutant. The compound is orally available and brain penetrant; however, it caused morphological changes in lungs of mice (all five of the tested animals developed enlarged type II pneumocytes upon MLi-2 dosing). Those changes nonetheless did not result in pulmonary or any other deficits and were completely reversible upon ceasing the treatment. Unfortunately, despite sustained LRRK2 kinase activity in the mouse brain, MLi-2 failed to slow or stop the progression of PD phenotype in the studied mouse model. Although MLi-2 did not make it to the clinic, it has become a valuable tool to study LRRK2 (patho)biology.

Another well characterized compound is GNE-7915. It was developed by Genentech in 2012 [85] based on a homology model of LRRK2. Highly soluble, very selective, potent, and brain penetrant, GNE-7915 (and a similar compound GNE-0877) was extensively tested in multiple species (mouse, rat, and cynomolgus monkey: [89]). Despite not showing any lung nor kidney pathology in rodents, all tested monkeys have developed abnormal accumulation of lamellar bodies in type II pneumocytes. Even though the effect was not accompanied by pulmonary deficits, this posed a serious safety concern for potential advancement to PD patients; therefore, clinical trials did not commence.

#### 3.1.1. Early Safety Concerns Regarding Kinase Inhibition

Despite the sheer amount of specific, brain penetrant, highly potent LRRK2 kinase inhibitors, almost all of them suffer from one drawback or another. An ideal inhibitor must not only be highly specific towards LRRK2, potent, and permeable though the blood–brain barrier, but, most importantly, must engage no other targets and display no (or only mild) side effects, need to be easily administered (oral administration is preferred) and must be safe in a prolonged use. This is especially important since PD patients will be taking the medicine for a long period of time (up to decades). Almost all of the developed up to date LRRK2 kinase inhibitors fail to meet some of the criteria. Medicinal chemistry efforts to produce better compounds are still ongoing, with DENALI leading the way by launching clinical trials for two of their compounds (vide infra).

The inhibitors that advanced to animal studies (MLi-2, GNE-7951, GNE-0877, PFE-360) resulted in kidney and lung phenotypes, raising major concerns regarding safety of LRRK2 kinase inhibition. However, the most recent MLi-2 safety re-evaluation study in macaques have shown that the lung phenotype is reversible and completely benign, not causing any respiratory issues [90]. Plausible explanation of the observed lung and kidney effects likely arises from the on-target engagement, as it has been shown that LRRK2 kinase inhibition leads to decreased LRRK2 levels due to its increased degradation via the ubiquitin pathway [91], which results in a phenotype similar to LRRK2 knock-out [83,91,92]. A recent study has proven that loss of the LRRK2 itself is neither toxic nor harmful in humans [93]; therefore, one may assume that the observed lung phenotype is innocuous. An alternative explanation for the observed on-target side effects of LRRK2 inhibition is stabilization of the closed conformation of the kinase domain, which leads to abnormal accumulation of LRRK2 on microtubules, where, in turn, it acts as a roadblock for both actin and dynein movement [27]. However, it must be noted here that this study was done in vitro, and accumulation of LRRK2 on microtubules so far has only been shown in cells that overexpress LRRK2. Therefore, it remains to be determined whether endogenous LRRK2 is also localized on microtubules and if (and how) this plays a role in PD and targeting of LRRK2.

#### 3.1.2. Rational Design of Improved ATP-Competitive Kinase Inhibitors

In the absence of high-resolution structures, the research has turned to molecular modeling and structural surrogate approach. Several models for the LRRK2 kinase domain have been developed, but they are mostly used in house and are not available for common use [85,94,95,96,97,98,99]. Some of them have been used for virtual screening for new inhibitors [96] and even de novo inhibitor design [94]. Although only two of the identified molecules have advanced to experimental use: GNE-7915 and PF-06447475 [82,85], they gave major insights into the amino acid residues in LRRK2 that are crucial for selective inhibition.

As for the structural surrogate approach (the rationale being to create a chimera of soluble kinase as a scaffold with the ATP-binding pocket properties of LRRK2), a number of structures with LRRK2-specific inhibitors bound in the ATP-binding pocket have been obtained [53,82,100,101], proving the efficacy of the approach; however, none of them have been used for computational design of improved LRRK2 kinase inhibitors so far.

Another interesting take on improving the existing LRRK2 inhibitors is approaching the issue from the perspective of the pharmacophore (e.g., properties of the ligand that are crucial for molecular recognition by the receptor; ATP-binding pocket of LRRK2 in this case). To the best of the author’s knowledge, only two such studies have been conducted [94,102]. Despite the great selectivity of the designed compound (25-fold better inhibition of the G2019S mutant over the WT LRRK2) [102], its experimental use is practically non-existent, and, to the best of author’s knowledge, sadly, the compound has not been studied in vivo.

Attempts have been made towards developing G2019S-specific inhibitors, to spare the WT LRRK2 activity in heterozygous patients. In a recent paper, Garofalo et al. have conducted a high-throughput screen for G2019S-specific inhibitors, and identified a single hit that led to a series of novel, potent, and highly selective inhibitors, reaching >300-fold selectivity in a cell-based assay on an endogenous LRRK2 [97]. Despite the compounds being poorly brain-penetrant, this study proves the concept of selective inhibition of a single point mutant, which paves the way for subsequent optimization and development of an improved inhibitor series.

#### 3.1.3. Allosteric LRRK2 Inhibitors Targeting the Kinase Domain

Targeting the ATP-binding site of any protein kinase is especially challenging due to high sequence and structural similarity between the kinases. Moreover, studies have shown that the most common G2019S mutation can be resilient to ATP-competitive inhibition [103]. A way to overcome these limitations is to target an allosteric site, if one could be identified in the LRRK2 kinase domain.

Identification of allosteric sites, however, is not a straightforward task, and is usually achieved by the means of a high-throughput screen, either an experimental or a computational one [104,105]. By “allosteric,” the authors mean here the classical definition, which is an effector site, distinct from the ATP-binding pocket, located within the kinase domain of LRRK2. However, achieving allosteric inhibition of the kinase domain by means of targeting other domains is also possible.

One of the physiologically active forms of vitamin B_12_, 5′-deoxyadenosylcobalamin (AdoCbl) was identified as an LRRK2 kinase inhibitor in a high-throughput screen [106]. Further characterization showed a moderate inhibitory effect in micromolar range on auto- and substrate phosphorylation, disruption of LRRK2 homodimerization, and neuron-loss rescue in simple animal models. AdoCbl was the only form of vitamin B_12_ that generated a response in vivo, in contrast with other vitamin B_12_ forms. The obtained results point at a mixed-type inhibition, which suggests, but does not equal, an allosteric mechanism. However, the AdoCbl concentrations that were used in these studies are very high, and, therefore, it is questionable whether AdoCbl can be useful for pharmacological treatment.

### 3.2. GTPase Modulators

Even though the exact mechanism of LRRK2 activation is still unclear, it seems apparent that the GTPase function is acting upstream from the kinase activity [107,108]. Therefore, instead of blocking the kinase domain, perturbing the Roc domain seems like a tempting strategy. 

From the literature, it is clear that the capacity to bind GTP and the capacity to hydrolyze it once it is bound are two separate events that have different mechanical outcomes on the kinase domain. The ability to bind GTP, and thus an intact Roc domain, is crucial for proper functioning of the kinase domain [107]. Artificial Roc mutants that cannot bind GTP (K1347A, T1348N) display no kinase activity; an artificial Roc mutant that can bind GTP, but cannot hydrolyze it (R1398L/T1343V) shows lowered kinase activity and neurite shortening; and, finally, an artificial mutant with normal GTP binding, but improved GTP hydrolysis (R1398L), shows normal kinase activity and rescues neurite shortening [34]. The GTP-locked mutant combined with G2019S mutation showed moderate rescue of neurite shortening. Therefore, one may conclude that a non-hydrolyzable GTP analog could benefit the patients with G2019S mutation, but not the others. Meanwhile, a molecule that improves GTP hydrolysis would be beneficial for all LRRK2 PD patients. As for complete abolishing of GTP binding, more data are needed to speculate, but it may be a third viable option [108]. In line with the benefits of stimulated GTPase activity, the protective LRRK2 variant (R1398H) is showing lower GTP binding but increased hydrolysis, and increased axon length compared with wild-type LRRK2 [70]. 

As a proof of concept, two GTP-competitive inhibitors were identified in a virtual database screening by Li et al. using the crystal structure of the Roc domain (PDB ID: 2ZEJ) as an input [72]. Those compounds, named 68 and 70, cause reduced GTP binding, kinase inhibition in vitro and in vivo alike, and rescue of neuronal degradation in a cell viability assay. Interestingly, both 68 and 70 did not affect GTP binding of the closest LRRK2 homolog, LRRK1, which shares 48% sequence similarity in the Roc domain. However, this study did not run any small G-protein specificity assays, while some cross-reactivity could be expected due to high similarity of the guanine nucleotide binding pocket. They showed that compound 68 was able to reduce LRRK2 phosphorylation in a mouse brain, showing its brain permeability. However, the dose that displayed the effect was 20 mg/kg, which is relatively high, while no effects could be seen at 10 mg/kg. An optimized compound, FX2149, displayed similar effects on GTP binding and kinase domain, and showed improved target engagement in the mouse brain, showing reduction of LRRK2 phosphorylation by 90% at a dose of 10 mg/kg [109]. All three Roc-specific inhibitors are presented in Table 3. This follow-up study, however, still showed no proof of specificity versus other small G-proteins. Compounds 68 and FX2149 were subsequently shown to rescue the impaired cargo transport along neurites in neuroblastoma SH-SY5Y cells, which further proves they are effective in LRRK2 inhibition [110]. Interestingly, in another follow-up study, it was found that compounds 68 and FX2149 increase aggresome formation and Lewy-body-like inclusions, as well as LRRK2 polyubiquitination via atypical K27 and K63 linkages [111]. Ubiquitin linkages via K27 are suggested to be a signal for protein aggregation [112], and K63 linkages are responsible for proteasome-independent processes [113]. Moreover, it remains to be determined whether protein aggregation is protective for neurons or not; however, Lewy bodies are generally believed to be neuroprotective. Interestingly, compound 68 was observed recently to lower the inflammatory response in immune cells [114]. Treatment with 68 reduced and attenuated the TNF-α release in LPS-treated lymphoblasts. This finding opens up a way to further investigate Roc domain inhibitors as potential anti-inflammatory agents in PD patients.

Together, these studies have proven that GTP-binding inhibitors could be an effective method of inhibiting LRRK2 and reducing neuroinflammation. Even though the exact mechanism of their action is unclear, it seems that altered GTP binding can influence the subcellular localization of LRRK2 via altered interactions with 14-3-3 proteins [115].

## 4. Downregulating LRRK2 Protein Levels

With a recent progress in precision medicine and gene therapy, an entirely different way to reduce the LRRK2 activity has emerged: reducing the amount of the produced protein by means of: a) modifying gene expression and splicing events or b) by actively degrading the already produced protein. To this end, several tools have been developed, which are mostly nucleic acid agents, such as: antisense oligonucleotides (ASOs), small interfering RNAs (siRNAs), short hairpin RNAs (shRNAs), microRNAs (miRNAs), splice-switching ASOs, and aptamers [8]. Next to those, a line of small molecules called proteolysis-targeting chimeras (PROTACs) is a very promising tool in neurodegenerative medicine [116]. So far, three reports of LRRK2-targeting nucleic acid agents have been published, and each of them adopts a different mechanism. 

The very early attempt at silencing LRRK2 expression by means of RNA interference was made by de Yñigo-Mojado et al. in 2011, who have identified two, allele-specific shRNAs that specifically target R1441G and R1441C alleles with 80% silencing efficiency [117]. The developed agents cleave the Lrrk2 mRNA by RNA-induced silencing complex, and can discriminate between WT and mutant Lrrk2 mRNA, thus not affecting the levels of the WT protein. As a proof of concept, these shRNAs were tested in human embryonal kidney cells (HEK 293FT), showing good selectivity over the WT gene (13.4 and 17.8-fold for each of the studied shRNAs) and a great silencing strength of 80%. This report shows, that, if needed, mutant forms of LRRK2 can be selectively silenced, which paved the way for further studies. Nonetheless, gauging the efficiency of this approach and potential side effects in neuronal cells or mice models would be highly desirable. 

Antisense oligonucleotides (ASOs) are single stranded, synthetic nucleic acids that bind to target mRNA by complementary base-pairing, which can result in mRNA degradation by RNAse H mechanism, among others [118]. Importantly, chemical modifications to the phosphodiester bond, sugar backbone, or other parts of the molecule can be introduced, in order to improve solubility, resistance to nucleases, and to improve other pharmacokinetic properties. Notably, ASOs can be delivered directly to the brain, without any carrier vesicles, by means of intracerebroventricular injection (ICV), and thus not affect any peripheral organs like lung and kidneys, as ASOs do not cross the blood–brain barrier [119]. Zhao et al. have developed two such ASOs, one of which is currently in clinical trials [120] (BIIB094, Table 4). They have found that both tested molecules produce a dose-dependent reduction of endogenous LRRK2 levels in brains of transgenic mice, without altering the levels of LRRK2 in other organs and not affecting LRRK1 levels. The developed ASOs reduced the formation of pathological α-synuclein inclusions, reduced the dopaminergic neuron cell loss, and reduced PD-related motor deficits. Importantly, long-term treatment with ASOs was well tolerated by the mice and did not result in any side-effects. Consistently, a recent study has shown that LRRK2 pLoF variants (loss-of-function of protein-coding genes) in a human cohort, results in reduced LRRK2 protein levels, but does not result in specific phenotype or disease [93]. Together, this strongly suggests that inhibition of LRRK2 expression will not result in major side-effects or symptoms.

These results show a great promise, with no side effects on the peripheral organs and their high selectivity; however, the main drawback is the invasive procedure of ICV injection. Alternatively, an Ommaya reservoir, a brain implant used for the aspiration of cerebrospinal fluid and drug delivery, could be used, but this also has a major impact on the quality of life of PD patients.

Another example of an interesting, new approach, are splice-switching ASOs developed by Korecka et al. [121]. The lrrk2 gene, coding for 51 exons, undergoes extensive splicing with seven various transcript forms and multiple splice variants [122]. It was proposed that editing out the exon 41, located within the kinase domain and encompassing the G2019 residue, would not only nullify the overactive mutation, but also lower the levels of WT protein. In addition, indeed, the obtained results show lower LRRK2 kinase activity measured as Rab10 phosphorylation, and normalized autophagic fluxes (measured as LC3B II/I ratio) upon ICV injection in transgenic mice [121]. A single ICV injection had long lasting effects: a decrease in LRRK2 protein levels three weeks after the procedure and decrease in Rab10 phosphorylation even after two months of the intervention. This observation is crucial from the prospective therapeutic point of view; as such, an invasive drug delivery method could be accepted every few months, if no other administration route is achieved.

An entirely different approach to downregulating protein levels is by means of targeted chemical degradation of translated protein, using PROTACs, that use the cell machinery to initiate degradation of the target protein. This approach has the advantage of degrading already formed protein aggregates and other targets that were previously deemed undruggable by conventional tools [116,123,124,125]. This promising idea in relation to LRRK2 was tested and a patent was recently published, where a set of PROTACs, that consist of a LRRK2 specific kinase inhibitor, a ligand that binds to the E3 ubiquitin ligase, and a linker connecting the two ligands, was generated. Studying this PROTACs in mouse embryonic fibroblast cell lines revealed decreased levels of LRRK2 and lowered S935 phosphorylation, showing the effectiveness of this approach (patent publication number WO 2020/081682 A1; [126]). 

Downregulating LRRK2 protein levels, either by means of degrading mRNA, altering splicing, or degrading the synthesized protein, is a very promising pharmacological approach. As genomic data showed, loss of LRRK2 function in humans is not associated with any specific phenotype or disease state [93], so even complete, uniform silencing of LRRK2 gene should yield desirable effects without observable toxicity or side effects. 

## 5. Conclusions and Outlook

From numerous studies, it is now clear that LRRK2 goes through a complex activation mechanism. The different PD-mutations all result in increased kinase activity and increased Rab-phosphorylation. Therefore, the most obvious way is direct kinase inhibition, which has attracted the most attention so far from both the academic and industrial community. Numerous ATP-competitive LRRK2-kinase inhibitors have been developed with some of them advancing to clinical trials, despite the initial safety concerns. Currently, three medicinal agents are in phase I clinical trials, the results of which are eagerly awaited (Table 4). In a press release from January 2020, Denali has announced that both tested compounds were well-tolerated, with little to no adverse side effects [127]. In a subsequent press release from August 2020, it was announced that both compounds have met the criteria to further advance in the clinical trials, with DNL151 being preferred due to more flexible dosing regimens [128]. An FDA application was also filed and approved for DNL151. Phase II for DNL151 is planned to commence in late 2021 [129]. The trial for BIIB094 is still actively recruiting participants, and no outcomes have been made public yet. It is estimated to complete in September 2023. A major challenge in clinical trials for drugs targeting PD is the lack of good biomarkers and setting the primary end-point, since most drugs do not modify the effect of the disease. However, LRRK2 activity is involved in the underlying process that plays a crucial role in both iPD and LRR2-mediated PD. Furthermore, reducing LRRK2 activity and/or levels has neuroprotective effects. Therefore, targeting LRRK2 has great potential as disease-modifying treatment [78].

Since the various PD-mutations have a different effect on the activation mechanism and there are still safety issues raised with the ATP-competitive kinase inhibitors, targeting other domains of LRRK2 than the kinase may prove to be therapeutically effective. Every step in the complex activation mechanism of LRRK2, including, but not limited to, the Roc domain, protein–protein interaction with the N- and C-terminal domains of LRRK2 (e.g., targeting binding of upstream Rab proteins), and/or dimerization, is a potential therapeutic target (Figure 3). The first Roc domain-targeting, GTP-competitive inhibitors have been developed. To identify new targeting surfaces and further develop these sorts of compounds for allosteric targeting of LRRK2 [130], further characterization of the LRRK2 activation mechanism and high-resolution structures will be of great importance. In this respect, the recently identified full-length structure of LRRK2 will be instrumental [31]. In addition to inhibiting LRRK2 activation, approaches that stimulate dephosphorylation of the major LRRK2 substrates, Rabs, could also be considered [131]. Then, finally, there are factors that could regulate the level of the LRRK2 protein itself, either by PROTACs or targeted gene therapy. 

Together, the data described in this review show that the LRRK2 field has seen great developments over the past decade and went from the lab to the clinic. A major question that remains to be answered in the upcoming years is whether the LRRK2 specific compounds would also benefit PD patients that do not carry LRRK2 mutations. The next big challenge in the field is the development of reliable biomarkers for accurate detection of LRRK2 activity and monitoring the progression of PD from the early stages. In this respect, antibody or mass spectrometry-based assays that can detect Rab10 phosphorylation in patients’ samples are being studied, as well as urinary proteome profiling, as non-invasive analytical methods [132,133,134].

## Figures and Tables

**Figure 1 biomolecules-11-01101-f001:**
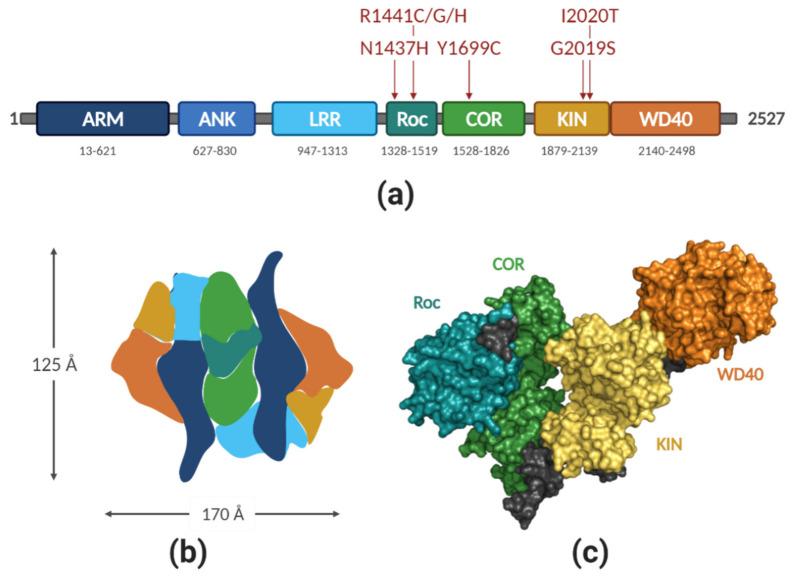
(**a**) Domain organization of LRRK2. ARM—Armadillo repeats, ANK—ankyrin repeats, LRR—leucin-rich repeats, Roc—Ras of complex proteins, COR—C-terminal of Roc, KIN—kinase, WD40—WD repeat domain. Red arrows above the protein indicate PD-causing mutations. Numbers below each domain indicate estimated domain boundaries; (**b**) approximate domain organization, based on the molecular model by Guaitoli et al. [28]. Color coding is the same as in (**a**); (**c**) structure of RocCOR-KIN-WD40 as solved by Deniston et al. [27] by cryo-EM with resolution of 3.5 Å. PDB code 6VNO; molecule visualized with PyMOL; figure created with BioRender.com (access date 2 April 2021).

**Figure 2 biomolecules-11-01101-f002:**
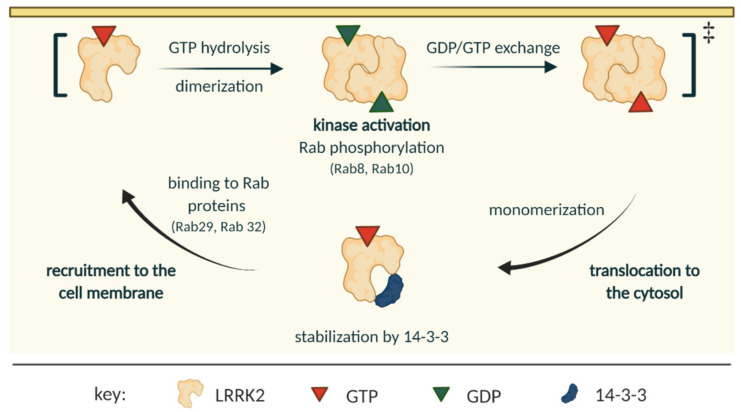
Current (simplified) model of the LRRK2 cycle. LRRK2 exists as monomers in the cytosol, where it is stabilized by 14-3-3 proteins. Upon binding to Rab proteins, LRRK2 is recruited to biological membranes, where the kinase domain of LRRK2 gets activated, the GTP hydrolyzes, and the protein dimerizes. The exact order in which these processes happen is unknown, but it is possible that they occur simultaneously. After the signaling output is achieved, GDP is exchanged to GTP, and the protein monomerizes and gets translocated to the cytosol. The figure is based on reference [52]. Association to microtubules was left out on purpose, as it remains to be experimentally proven to be true for endogenous LRRK2; figure created with BioRender.com (access date 2 April 2021).

**Figure 3 biomolecules-11-01101-f003:**
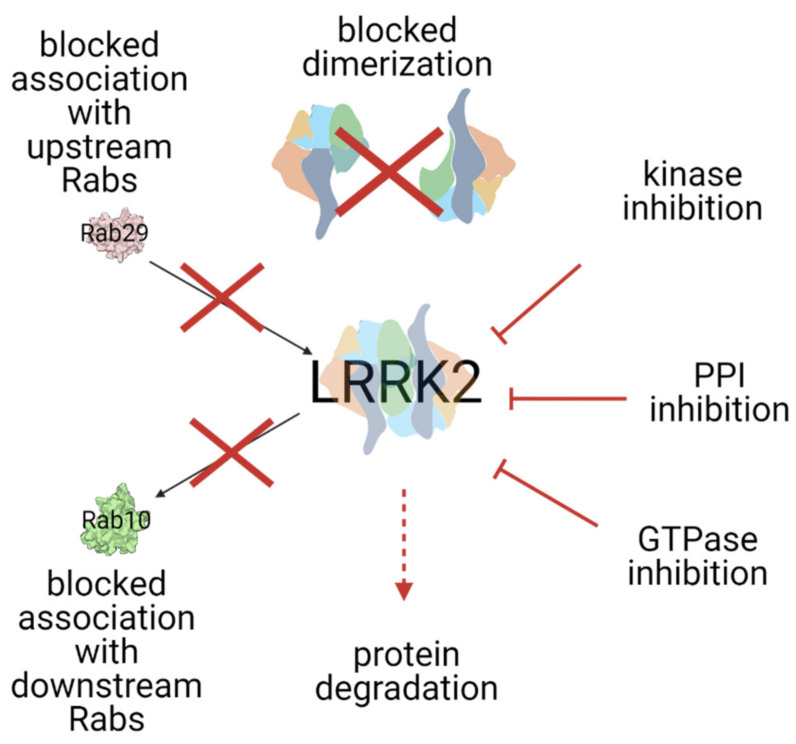
Summarized LRRK2 targeting strategies. See the text for a detailed explanation of each strategy.

**Table 1 biomolecules-11-01101-t001:** Most common, PD-causing LRRK2 mutations and their effect on the protein activity.

Mutation	Domain	Effect on the Kinase	Effect on the GTPase	Probable Mechanism
R1441G	Roc	no effect [65] 11–15× ↑ Rab phosph. [66] 2.5× ↑ autophosph. [67]	↑ GTP binding, ↓ GTP hydrolysis	loss of positive charge that impairs dimerization [68]
R1441C	Roc	4–6× ↑ Rab phosph. [66] 3× ↑ autophosph. [67]	↑ GTP binding, ↓ GTP hydrolysis [69]	↓ thermodynamic stability of Roc domain [70]; loss of positive charge that impairs dimerization [68]
R1441H	Roc	9–10× ↑ Rab phosph. [66]	↑ GTP binding, ↓ GTP hydrolysis [65]	loss of positive charge that impairs dimerization, alteration to tertiary structure of Roc domain [68]
Y1699C	COR	14–18× ↑ Rab phosph. [66] no effect on autophosph. [67]	↑ GTP binding, ↓ GTP hydrolysis [71]	alteration of electrostatic surface [68]
G2019S	kinase	2–3× ↑ [66] 3.5× ↑ autophosph. [67]	no effect [62] ↑ GTP binding [72]	DYG loop stabilized in active conformation by hydrogen bond [49,68,73]
I2020T	kinase	no effect in vitro [74] 6–7× ↑ Rab phosph. [66] 2× ↑ autophosph. [67]	↑ GTPase activity [74]	affected stability, kinase stabilized in an inactive conformation by hydrogen bond [68,73]

phosph.—phosphorylation; autophosph.—autophosphorylation; ↑—increased, ↓—decreased.

**Table 2 biomolecules-11-01101-t002:** Most used, orthosteric LRRK2 kinase inhibitors and their selected properties.

Compound Name	Chemical Structure	LRRK2 IC_50_ [nM]	Brain Permeability	Reference
WT	G2019S
MLi-2	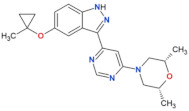	0.8	0.76	yes	[80,81]
PF-06447475	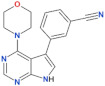	3	11	yes	[82]
PF-06685360	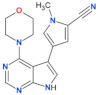	2.3	n.d.	yes	[83]
GNE-0877	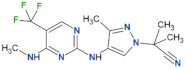	3	n.d.	yes	[84]
GNE-7915	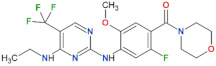	9	n.d.	yes	[85]
GSK2578215A	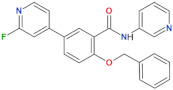	10.9	8.9	yes	[86]
HG-10-102-1	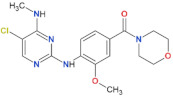	20.3	3.2	yes	[87]
LRRK2-IN-1	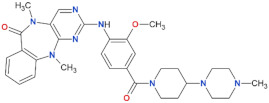	13.0	6.0	no	[88]

n.d.—no data; NHP—non-human primate.

**Table 3 biomolecules-11-01101-t003:** LRRK2 GTPase inhibitors developed to date.

Compound Name	Chemical Structure	Brain Permeability	Reference
68	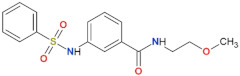	yes	[72]
70	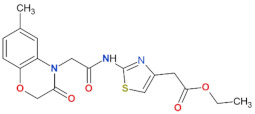	n.a.	[72]
FX2149	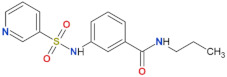	yes	[109]

n.a.—not available (compound insoluble).

**Table 4 biomolecules-11-01101-t004:** LRRK2 therapeutics that are currently in clinical trials.

Clinical Trial	Compound Name	Compound Type	Phase	Funding Body
NCT03710707	DNL201	kinase inhibitor	Phase Ib (completed)	Denali Therapeutics Inc. (South San Francisco, CA, USA)
NCT04056689	DNL151	kinase inhibitor	Phase Ib (in progress)	Denali Therapeutics Inc. (South San Francisco, CA, USA) and Biogen (Cambridge, MA, USA)
NCT03976349	BIIB094	ASO	Phase I (recruiting)	Biogen (Cambridge, MA, USA)

## Data Availability

Not applicable.

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
