# Peer review of "LRRK2 Targeting Strategies as Potential Treatment of Parkinson’s Disease"

_biomolecules, 2021, doi:10.3390/biom11081101_

Round 1
Reviewer 1 Report
This is an interesting topic and the manuscript is of broad interest in the field. Overall, the authors nicely review the therapeutic strategies that are currently being pursued to tackle Parkinson’s disease via inhibition of LRRK2 activity. However, I recommend the authors to have the manuscript reviewed for clarity by a native speaker of English. It is my personal opinion that the writing style should be further improved and become more elegant and attractive to the reader.
Some minor concerns are stated below:
The authors frequently refer to other bibliographic references to minimize further explanations, such as in Lines 39-41: “Epidemiological studies have shown that there is strong genetic correlation, with mutations in five genes being the major culprit: SNCA (coding for α-synuclein), LRRK2, VPS35, PRKN, DJ-1, and PINK1 (see the reference [7] for more details (…)”. This type of construction could be avoided. Also, in the same sentence, the proteins encoded by VPS35, PRKN, DJ-1, and PINK1 genes should be mentioned.
In Lines 200-202 the authors cite 8 Reviews, which is somehow exaggerated. In addition, it is always preferable to cite the original papers instead of Reviews. Exception made for doi.org/10.1038/s41582-019-0301-2 that brilliantly summarizes the challenges of LRRK2-targeted therapies and clinical trials.
Author Response
This is an interesting topic and the manuscript is of broad interest in the field. Overall, the authors nicely review the therapeutic strategies that are currently being pursued to tackle Parkinson’s disease via inhibition of LRRK2 activity. However, I recommend the authors to have the manuscript reviewed for clarity by a native speaker of English. It is my personal opinion that the writing style should be further improved and become more elegant and attractive to the reader.
We thank the reviewer for carefully reading our manuscript and have addressed all the comments raised.
Some minor concerns are stated below:
The authors frequently refer to other bibliographic references to minimize further explanations, such as in Lines 39-41: “Epidemiological studies have shown that there is strong genetic correlation, with mutations in five genes being the major culprit: SNCA (coding for α-synuclein), LRRK2, VPS35, PRKN, DJ-1, and PINK1 (see the reference [7] for more details (…)”. This type of construction could be avoided. Also, in the same sentence, the proteins encoded by VPS35, PRKN, DJ-1, and PINK1 genes should be mentioned.
We thank the reviewer for this suggestion and changes this is the revised manuscript.
In Lines 200-202 the authors cite 8 Reviews, which is somehow exaggerated. In addition, it is always preferable to cite the original papers instead of Reviews. Exception made for doi.org/10.1038/s41582-019-0301-2 that brilliantly summarizes the challenges of LRRK2-targeted therapies and clinical trials.
We have reduced the amount of references and only referred to doi.org/10.1038/s41582-019-0301-2.
Reviewer 2 Report
In the review by D. N. Wojewska and A. Kortholt, the authors summarize current approaches to LRRK2 targeting drug development. This is a very thoughtful, comprehensive and well-structured review. The authors explained mechanisms in a clear to the reader way. I have only minor stylistic comments.
-genes should be in Italic.
-Please double check abbreviations. (i.e. Parkinson’s Disease [33]. on page 6 line 193)
-I would suggest change “most used” term in “Modes of LRRK2 inhibition” section to something similar to “well studied”. As it is only used in preclinical stages for research and not used in clinical practice.
-“ Another very promising compound, GNE-7915” how Is it promising with all the safety issues? It was not even tried in PD, not promising at all. Please rephrase.
Author Response
In the review by D. N. Wojewska and A. Kortholt, the authors summarize current approaches to LRRK2 targeting drug development. This is a very thoughtful, comprehensive and well-structured review. The authors explained mechanisms in a clear to the reader way. I have only minor stylistic comments.
We thank the reviewer for carefully reading our manuscript and have addressed all the comments raised.
-genes should be in Italic.
We have corrected this in the revised manuscript
-Please double check abbreviations. (i.e. Parkinson’s Disease [33]. on page 6 line 193)
This has been corrected in the revised manuscript.
-I would suggest change “most used” term in “Modes of LRRK2 inhibition” section to something similar to “well studied”. As it is only used in preclinical stages for research and not used in clinical practice.
We have changed this in the revised version of the manuscript.
-“ Another very promising compound, GNE-7915” how Is it promising with all the safety issues? It was not even tried in PD, not promising at all. Please rephrase.
We agree with the reviewer and have changed this in the revised manuscript
Reviewer 3 Report
Pls refer to the attached file for my comments on this article.

Author Response
The review article aimed to provide a comprehensive overview on the protein
structure and biological function of Leucine-rich repeat kinase 2 (LRRK2), which gene
contributes to the most common genetic cause of autosomal dominant Parkinson’s
disease in the world. Moreover, the authors review the current development of
LRRK2 kinase inhibitors as the therapeutic strategy in the treatment of Parkinson’s
disease and preventing the progression of the disabled neurodegenerative disorders.
The article is well written in the organization and describes comprehensively the
various modes of LRRK2 inhibition including the kinase inhibitors and GTPase
modulators.
We thank the reviewer for carefully reading our manuscript and have all the comments raised.
- In the Introduction, the authors described there are 5 major genes including
SNCA, LRRK2, VPS35, PRKN, DJ1 and PINK contributing to the genetic causes in
hereditary Parkinson’s disease. However, there are great racial difference of the
genetic causes in different populations, for example, SNCA and DJ1 are very rare in
Asian. Mutation or variants of GBA, encoding the lysosomal enzyme
glucocerebrosidase (GCase) plays a major role in the genetic cause of risk factor in
association with Parkinson’s disease in Ashkenazi Jewish. Therefore, in this
paragraph, I would like to suggest authors to describe more about the racial
difference in the genetic causes of hereditary Parkinson’s disease.
We thank the reviewer for pointing this out and have now addressed this in more detail on page 2 of the revised manuscript.
- In section 2.3 LRRK2 activation in Parkinson’s disease. It has been known that
several points mutations in Roc COR and Kinase domains are segregated with the
autosomal dominant Parkinson’s disease (PD) and may be pathogenic to the
underlying mechanism of neuronal death. However, the authors did not describe
the clear evidence of LRRK2 kinase or GTPase dysfunction in sporadic Parkinson’s
disease. The current LRRK2 inhibitors are designed for the treatment of sporadic
Parkinson’s disease, not limited to patients with LRRK2 gene mutations or variants in
the functional domains. The authors are advised to provide more evidence to the
association of LRRK2 kinase dysfunction in idiopathic PD or the benefit of decreased
kinase activity in the underlying pathogenesis of dopaminergic neuronal death in
idiopathic PD.
We thank the reviewer for this good suggestion and have now provided evidence for this on page 6 of the revised manuscript.
- In section 4. Downregulation of LRRK2 levels by antisense oligonucleotides
(ASOs). LRRK2 plays several important physiological functions including the
endosome-lysosomal function and vesicle trafficking in the neurons. The authors are
advised to describe the possible concern of the safety with decreased LRRK2 level in
the normal brain function, instead of only focusing on the reduction of α-synuclein
aggregation and motor deficit in both animal and human studies.
We thank the author for the suggestion and have added this on page 11 of the revised manuscript.
- In perspective of the clinical trial with LRRK2 kinas inhibitors in the treatment of
idiopathic Parkinson’s disease, the authors are advised to provide their commons on
the selection of the primary end-point in the clinical trials listed in Table 4. One of
the current unmet need in the treatment of PD is lack of drugs providing disease
modifying effect. What is the therapeutic strategy of the LRRK2 kinase inhibitor or
ASOs in the symptomatic or neuroprotective treatment in idiopathic PD?
We thank the reviewer for this suggestion, we have no briefly addressed this on page 14 of the revised manuscript.